# Cognitive decline in patients with prostate cancer: study protocol of a prospective cohort, NEON-PC

Natalia Araujo,[1] Samantha Morais,[1,2] Ana Rute Costa,[1] Raquel Braga,[1,3] Ana Filipa Carneiro,[4] Vitor Tedim Cruz,[1,5] Luis Ruano,[1,6] Jorge Oliveira,[7] Luis Pacheco Figueiredo,[8,9] Susana Pereira,[1,10] Nuno Lunet 🔾 [1,2]

For numbered affiliations see end of article.

**Correspondence to**
Dr Nuno Lunet;
nlunet@med.up.pt

## ABSTRACT

**Introduction** Prostate cancer is the most prevalent oncological disease among men in industrialised countries. Despite the high survival rates, treatments are often associated with adverse effects, including metabolic and cardiovascular complications, sexual dysfunction and, to a lesser extent, cognitive decline. This study was primarily designed to evaluate the trajectories of cognitive performance in patients with prostate cancer, and to quantify the impact of the disease and its treatments on the occurrence of cognitive decline.

**Methods** Participants will be recruited from two main hospitals providing care to approximately half of the patients with prostate cancer in Northern Portugal (Portuguese Institute of Oncology of Porto and São João Hospital Centre), and will comprise a cohort of recently diagnosed patients with prostate cancer proposed for different treatment plans, including: (1) radical prostatectomy; (2) brachytherapy and/or radiotherapy; (3) radiotherapy in combination with androgen deprivation therapy and (4) androgen deprivation therapy (with or without chemotherapy). Recruitment began in February 2018 and is expected to continue until the first semester of 2021. Follow-up evaluations will be conducted at 1, 3, 5, 7 and 10 years. Sociodemographic, behavioural and clinical characteristics, anxiety and depression, health literacy, health status, quality of life, and sleep quality will be assessed. Blood pressure and anthropometrics will be measured, and a fasting blood sample will be collected. Participants' cognitive performance will be evaluated before treatments and throughout follow-up (Montreal Cognitive Assessment and Cube Test as well as Brain on Track for remote monitoring). All participants suspected of cognitive impairment will undergo neuropsychological tests and clinical observation by a neurologist.

**Ethics and dissemination** The study was approved by the Ethics Committee of the hospitals involved. All participants will provide written informed consent, and study procedures will be developed to ensure data protection and confidentiality. Results will be disseminated through publication in peer-reviewed journals and presentation in scientific meetings.

## Strengths and limitations of this study

► This protocol describes a prospective cohort study of patients with prostate cancer, expected to reflect the contemporary patterns of diagnosis and treatment in developed countries.

► Cognitive impairment will be characterised regarding its severity and possible aetiologies through neuropsychological and clinical evaluations.

► Short-term and long-term effects as well as mediators of the effect of androgen deprivation therapy on cognitive performance will be analysed.

► A longitudinal remote monitoring tool of cognitive function will be used, in addition to state-of-the-art methods, which allows for more frequent standardised evaluations, while reducing learning effects of repeated measurements.

► Only a measure of overall cognitive function will be obtained from all participants and multiple cognitive domains will only be evaluated in patients with probable cognitive impairment.

death from cancer among men, with nearly 1.3 million new cases and 359 thousand deaths estimated in 2018 worldwide.[1] In recent decades, prostate cancer incidence has been heavily influenced by diagnoses following prostate-specific antigen testing of asymptomatic individuals and by the detection of latent cancer in tissue removed during prostatectomy or autopsy.[1] At the same time, prostate cancer mortality has been decreasing in many countries, which has been linked to earlier diagnosis because of extensive use of prostate-specific antigen screening, and improved treatment including radical prostatectomy, hormonal therapy and radiation therapy.[2 3] Increases in prostate cancer survival[4] require a comprehensive assessment of the burden of cancer, due to the disease, treatment and sequelae.[5 6]

Androgen deprivation therapy is used in the treatment of approximately half of all patients with prostate cancer,[7 8] and it may

## INTRODUCTION

Prostate cancer is the second most common neoplasm and the fifth-leading cause of

last from 6 to 36 months, on an intermittent basis or continue indefinitely.[9] The use of androgen deprivation therapy and its impact on cognitive function has been assessed, both in prospective studies evaluating cognitive performance using neuropsychological tests and in large retrospective studies reporting the risk of dementia or of Alzheimer's disease in patients with prostate cancer according to androgen deprivation therapy exposure.[10–12] However, methodological heterogeneity does not allow for the direct comparison of results, and shortcomings of the study designs, including small sample sizes, short follow-up periods or limited quality of information on cognitive status, as well as residual confounding, preclude more robust conclusions on this topic.[10] Also, in addition to the possible direct effect of androgen deprivation therapy on cognitive function due to the drop in serum testosterone and its biological activity in certain areas of the brain,[13] hormonal changes may also cause metabolic alterations,[14] with an increase in cardiovascular risk factors, such as an increase in insulin resistance, serum cholesterol and triglycerides or anaemia,[15] which in turn are associated to cognitive decline.[16–19] This possible indirect effect may take longer to manifest in the brain than the direct decrease in testosterone serum levels, and it may be related to the development of dementia. The potential mediator effect of these biochemical and haematological parameters has not been studied. Prospective investigations including an accurate characterisation of the cognitive performance of patients with prostate cancer proposed for different types of treatment, and analyses accounting for distinct causal pathways may contribute to a better understanding of the effects of prostate cancer and its treatments on cognitive decline.

Therefore, this project primarily aims to understand the impact of androgen deprivation therapy on the cognitive performance of patients with prostate cancer in Northern Portugal. The main specific objectives are as follows:

1. To describe the trajectories of cognitive performance over time (up to 10 years) in patients with prostate cancer under different treatments and, in comparison to the general population, by using the Montreal Cognitive Assessment tool, the Cube Test and Brain on Track. The relation between patients' characteristics, cancer treatments and different cognitive trajectories will also be assessed.
2. To quantify the association between androgen deprivation therapy and cognitive decline, in the short term and in the long term.
3. To assess the role of metabolic syndrome and anaemia as possible mediators of the androgen deprivation therapy effect on cognitive performance.

## METHODS AND ANALYSIS

We describe a prospective cohort study that will evaluate patients with prostate cancer selected among those being treated at the two largest hospitals providing cancer care in the North of Portugal, which attend half of the patients with prostate cancer in this region. Recruitment started in February 2018 and is ongoing. We expect to complete it in the first semester of 2021.

### Eligibility criteria

Eligible participants are those with a recent diagnosis of prostate cancer and being initially proposed for radical prostatectomy (group 1), brachytherapy or radiotherapy (group 2), radiotherapy in combination with androgen deprivation therapy (group 3), or androgen deprivation therapy with or without chemotherapy (group 4), and prostate cancer survivors never treated with androgen deprivation therapy before, who present with a recurrence of the disease to be treated with androgen deprivation therapy, with or without chemotherapy (group 5).

Participants who had a previous chemotherapy or radiotherapy treatment for another primary cancer, or a diagnosis of a psychiatric or a neurological condition impairing cognitive function before the prostate cancer diagnosis, or being unable to understand the purpose of the study or to collaborate will be excluded, as well as those expected to receive cancer treatments outside the Portuguese Institute of Oncology of Porto or the São João Hospital Centre.

### Participants' recruitment and follow-up

Patients with prostate cancer will be consecutively recruited at the Portuguese Institute of Oncology of Porto and the São João Hospital Centre, from February 2018 to the first semester of 2021. Participants will be evaluated at baseline (before any treatment for recently diagnosed patients or before androgen deprivation therapy for patients with a recurrence of the disease), and at 1, 3, 5, 7 and 10 years after enrolment, as depicted in figure 1.

### Data collected from medical records

Clinical characteristics, including comorbidities, medication and cancer treatment (including all drugs used for systemic treatment of prostate cancer, either at initial or follow-up treatments and duration), as well as prognostic and treatment response biomarkers will be systematically collected by medical doctors from the patients' medical records. Prostate cancer staging based on tumor (T), nodes (N) and metastases (M) (TNM stages) will be in accordance with the American Joint Committee on Cancer TNM system classification[20] and risk stratification according to the National Comprehensive Cancer Network (www.nccn.org).

### Questionnaire evaluation

Data on sociodemographic (birth date, address, marital status, education, occupation), lifestyle and dietary characteristics (smoking and alcohol consumption, and intake of fruits and vegetables, physical activity and sedentary behaviours) will be collected through questionnaires applied by a trained interviewer. Anxiety and depression,[21 22] sleep quality,[23 24] quality of life and health status,[25–28] and health literacy[29 30] will be evaluated through self-administered questionnaires validated for

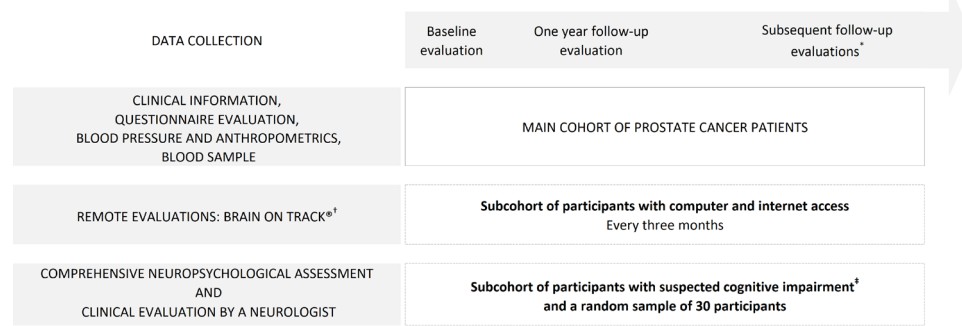

**Figure 1** Study design, and timing of baseline and follow-up evaluations in the main cohort and the subcohort of participants with suspected cognitive impairment. *Subsequent follow-up evaluations will be at 3, 5, 7 and 10 years after the baseline evaluation. †The Brain on Track evaluation will be conducted every 3 months. ‡Only participants who score below 1.5 SD of age-adjusted and education-adjusted cut-offs on the Montreal Cognitive Assessment during each evaluation (baseline, and 1, 3, 5, 7 and 10 years of follow-up) and a random sample of 30 participants will be invited for a neuropsychological evaluation where a battery of cognitive tests will be applied. The type of cognitive impairment will be classified through a clinical evaluation performed by a neurologist.

the Portuguese population, and are described in detail in table 1.

## Blood pressure

Blood pressure will be measured with a digital blood pressure monitor (Omron M6). Participants will be asked to remain seated, with the right arm and back supported and feet firmly on the floor, and to abstain from speaking during the entire procedure. The cuff will be placed on the right arm so the bottom margin is approximately 2–3 cm above the antecubital fossa. A larger or a smaller cuff will be used as necessary to fit the arm of the participant. Three measurements with 1 min intervals will be registered.

## Anthropometrics

Weight and height will be measured with participants in light clothes and no shoes, and registered to the nearest kilogram and centimeter, respectively, using a digital column scale (Seca 799). Waist and hip perimeters will be measured using a non-elastic measuring tape (Seca 211) with participants standing, with feet slightly apart and the arms relaxed along the body; waist perimeter will be measured at half the distance between the last rib and the iliac crest. Hip perimeter will be measured with participants in the same position, with the measuring tape placed at the widest part of the hip below the iliac crest. Both waist and hip circumferences will be registered to the nearest millimeter. Most measurements are expected to be performed in the morning.

## Blood sample

A fasting blood sample (at least 12 hours) will be collected by the hospitals' nurses using venous puncture, and blood samples will be centrifuged at 3000 rpm for 10 min to obtain plasma and serum, within 30–60 min. Total cholesterol, high density lipoprotein cholesterol, triglycerides, glycaemia, glycated haemoglobin and haemoglobin will be analysed. Plasma and serum samples will be stored

in small aliquots at −80°C until the end of the study (10 years).

## Cognitive function evaluation

Cognitive function will be evaluated using the Montreal Cognitive Assessment[31 32] and the Cube Test,[33] at baseline and at each of the subsequent follow-up evaluations, and with a web-based tool for remote longitudinal assessment (Brain on Track),[34] every 3 months for a period of up to 10 years.

Participants suspected of cognitive impairment will undergo a comprehensive neuropsychological assessment that will allow specific domains of cognitive function to be analysed; the battery of tests is described in table 2. Additionally, those with confirmed cognitive impairment will undergo a clinical evaluation by a neurologist.

### The Montreal Cognitive Assessment

The Montreal Cognitive Assessment is a pen-and-paper screening test, developed to detect mild cognitive impairment. It assesses eight cognitive domains (visuospatial ability, executive function, attention, concentration, working memory, language, verbal memory and orientation), generating a total score ranging from 0 to 30.[31] The translated, culturally adapted and validated version of the Montreal Cognitive Assessment for the Portuguese population[35] will be used, and the performance of participants will be classified as probable cognitive impairment when the score is 1.5 SD below the mean of age-based and education-based group distribution from published normative data.[32]

### The Cube Test

The Cube Test will be used as a rapid cognitive screening tool, which can be applied to illiterate participants, or those with low educational levels, language or hearing deficits.[33] The Cube Test is easy to apply, and the simple instructions and scoring procedures contribute for standardised use. The test is based on the time spent in

**Table 1** Description of the instruments used for the evaluation of all patients

| Instrument | Description | Domains/subscales | Score |
|---|---|---|---|
| MoCA[31 35] | Test for the rapid screening of mild cognitive impairment—an intermediate clinical state between normal cognitive ageing and dementia. | Attention and concentration; executive functions; memory; language; visuoconstructional skills; calculations; orientation. | Range: 0–30<br>Higher scores represent better cognitive performance. |
| Cube Test[33] | A two-task cognitive screening tool that consists in completing a 3D cube from six pieces (task 1) and remembering the position of the six pieces on a grid with 25 squares measuring eight by eight centimeters from a previously shown scheme (task 2). | Visuoconstructional skills; executive function; processing speed; delayed memory. | Time to construct the first vertex and to complete cube and the number of pieces correctly assembled in up to 6 min (task 1); number of pieces correctly positioned on the grid (task 2). |
| Brain on Track[34] | A self-administered computerised test intended for longitudinal cognitive testing that includes eight subtests. | Attention; memory; executive functions; language; calculation; constructive ability; visuospatial processing. | Range: virtually unlimited (maximum number of correct answers in a fixed time)<br>Higher scores represent better cognitive performance.<br>Scores falling below an expected performance threshold for each age/education group; a pattern of decline in individual performance. |
| HADS[21 22] | Scale with 14 questions assessing anxiety and emotional distress among patients during the previous week. | Depression; anxiety. | Range (for each subscale): 0 to 21<br>Scores greater than or equal to 11 represent a case of anxiety or depression, as applicable. |
| PSQI[23 24] | Index with 18 questions assessing sleep quality and disturbances during the previous month. | Subjective sleep quality; sleep latency; duration of sleep; habitual sleep efficiency; sleep disorders; use of medications for sleep; daytime dysfunction. | Range: 0–21<br>Scores greater than five indicate poor sleep quality. |
| QLQ-C30[25 26] | Scale with 30 questions assessing quality of life in patients with cancer during the previous week. | Global health status.<br>Functional scales: physical functioning; role functioning; emotional functioning; cognitive functioning; social functioning.<br>Symptom scales/items: fatigue; nausea and vomiting; pain; dyspnoea; insomnia; appetite loss; constipation; diarrhoea; financial difficulties. | Range (scales and single-item): 0–100<br>Higher scores for the global health status and for a functional scale represent a healthy level of quality of life and functioning, respectively.<br>Higher scores for a symptom scale/item represents a higher level of symptomatology/ problems. |
| QLQ-PR25[27] | Specific Prostate Cancer Scale with 25 questions assessing quality of life in patients with prostate cancer during the previous week and the last 4 weeks. | Functional scales: sexual activity; sexual functioning.<br>Symptom scales: urinary symptoms; bowel symptoms; hormonal treatment-related symptoms; incontinence aid. | Range (scales and single-item): 0 to 100<br>Higher scores for a function scale/item reflect a healthy level of functioning.<br>Higher scores for a symptom scale/item reflect a higher level of symptomatology/problems. |
| EQ-5D-5L[28] | A measure of health-related quality of life with five questions and a Visual Analogue Scale. | Mobility, self-care, usual activities, pain/discomfort; anxiety/depression and a visual analogue scale. | A total of 3125 possible health states are defined to describe the patient's health state. Each state is referred to in terms of a 5-digit code.<br>Vertical Visual Analogue Scale<br>Range: 0–100<br>Higher scores reflect 'The best health you can imagine' and lower scores reflect 'The worst health you can imagine'. |
| METER[29 30] | A measure of health literary including 40 words and 30 non-words. | 40 words and 30 non-words. | Range: 0–40 and 0–30<br>Adequate health literacy is defined as scoring at least 35/40 in words and 18/30 in non-words. |

3D, three dimensions; EQ-5D-5L, Measure of health-related quality of life of the EuroQol Group; HADS, Hospital Anxiety and Depression Scale; METER, Medical Term Recognition Test; MoCA, Montreal Cognitive Assessment; PSQI, Pittsburgh Sleep Quality Index; QLQ-C30, Quality of Life Questionnaire of the European Organisation for Research and Treatment of Cancer; QLQ-PR25, Prostate cancer-specific module of the Quality of Life Questionnaire of the European Organisation for Research and Treatment of Cancer.

assembling the six faces of a 3D cube and then, the correct placement of the six faces of the 3D cube on a grid with 25 squares measuring eight by eight centimeters. The Cube Test assesses visuoconstructive, visuospatial and executive functions, visuospatial working memory, information processing speed, incidental learning, motor processing speed and manual dexterity.

### Brain on Track

The Brain on Track test will be used for the remote evaluation of changes in cognitive function. This is a computerised cognitive monitoring test, which was developed and validated in the Portuguese population, showing good internal consistency, discriminative ability and reliability.[34 36] Brain on Track evaluates different cognitive

**Table 2** Description of the battery of instruments used for the comprehensive neuropsychological assessment*

| Instrument | Description | Cognitive domains/function | Score |
|---|---|---|---|
| SMC[45 46] | A 10-item scale regarding subjective memory complaints. | Subjective memory. | Range: 0–21<br><br>Higher scores reflect maximal memory complaints. |
| Phonemic Verbal Fluency[47 48] | A test consisting of three trials of 1 min each where participants are asked to produce orally as many words as possible beginning with a specific letter. | Executive function; language; semantic memory. | The total trial score corresponds to the no of words correctly produced within 1 min. The total test score corresponds to the sum of the three trials.<br><br>Higher scores correspond to better performance. |
| 18-point CDT[49–51] | An 18-point clock-drawing scoring system where participants are asked to draw a big circle and put the numbers of the clock, and then they were asked to indicate the time as '10 past 11'. | Visuospatial; executive function. | Range: 0–18<br><br>Scoring system with three main components: (1) assessment of circle integrity (two points); (2) number placement and sequencing (six points) and (3) placement and size of the hands (six points). Additionally, there are two points for representation of the clock's centre and two points for general gestalt. |
| TMT[52 53] | Part A: participants are asked to draw lines to connect 25 randomly positioned numbered circles in numeric order as quickly as possible.<br><br>Part B: participants are asked to draw lines to connect circles in numeric and alphabetic order as quickly as possible, alternating between numbers and letters (progressively up to number 13). | Part A: attention; visual scanning and speed of eye-hand coordination and information processing.<br><br>Part B: working memory and executive functions; particularly, the ability to switch between sets of stimuli. | Direct measures of performance: time (seconds) to complete part A and part B, and performance errors during part A and part B.<br><br>Derived scores: difference score (B–A), ratio score (B/A), proportion score (B–A/A), sum score (A+B), and multiplication score (A×B/100).<br><br>Lower raw scores and higher adjusted scores correspond to better performance. |
| WMS-III[54 55] | Evaluates memory and attention functions using both auditory and visual stimuli.<br><br>A test composed of 17 subtests designed to measure different memory functions in a person with the aim of detecting and discriminating between subcortical vascular dementia and Alzheimer's disease.<br><br>Subtests used: Logical Memory I, II; Visual Reproduction I, II; Digit Span. | Verbal and visual memories; working memory. | Range:<br>► Immediate recall: 0–50<br>► Delayed recall: 0–50<br>► Auditory recognition: 0–30<br>► Visual reproduction: 0–104<br>► Digit span: 0–30<br><br>Higher scores correspond to better performance. |
| WAIS-III[56 57] | Measures intelligence and cognitive ability in adults and older adolescents.<br><br>Subtests used: Digit-Symbol-Coding, which consists of digit-symbol pairs followed by a list of digits and under each digit participants write down the corresponding symbol as fast as possible; and Symbol Search, in which, participants are asked to look at two groups of symbols and to indicate if any of the symbols of the first group are present in the other group. | Attention/concentration; executive function (sequencing); motor function; processing speed. | The number of correct symbols within the allowed time (120 s) is measured. |
| Stroop Test[58] [59] | Assesses the ability to inhibit cognitive interference, which occurs when the processing of a stimulus feature affects the simultaneous processing of another attribute of the same stimulus.<br><br>This test has three trials: (1) the participant is required to read the colour names printed in black ink as quickly as possible; (2) the participant is required to name the colour of coloured dots as quickly as possible; (3) the participant is required to name the colour of the ink of the colour name words (the colour name does not match the colour of the ink). | Executive functions (inhibitory control); selective attention. | Scores for each trial indicate the number of correct responses. An interference score can be generated that quantifies the participant's ability to inhibit the inappropriate response of reading the colour name as opposed to the colour of the ink used to print the colour name in the third trial. |

Continued

**Table 2** Continued

| Instrument | Description | Cognitive domains/ function | Score |
|---|---|---|---|
| Token Test-short form[60] | A test designed to assess the comprehension of commands that vary in degree of linguistic difficulty but which are relatively independent of defects in other aspects of intellectual capacity such as memory and vocabulary. The test consists of six subsections that represent different levels of linguistic difficulty. The participant is presented with tokens of different shapes (ie, circles, squares, triangles), sizes, and colours, and is required to perform certain acts with the tokens, such as point to selected tokens, touch them, pick them up and place one token on top of another. | Attention and vigilance; verbal functions. | Range: 0–36 Higher scores correspond to better performance. |
| SDMT[61] | A quick screening test for organic cerebral dysfunction. The test involves a simple substitution task that can be easily performed: using a reference key, the participant has 90 s to pair specific numbers with given geometric figures. | Organic cerebral dysfunction. | Individuals with cerebral dysfunction perform poorly. |
| TeLPI[62] | A Portuguese irregular word reading test using 46 irregular, infrequent Portuguese words designed to assess premorbid intelligence. | Premorbid IQ: full scale IQ; Verbal IQ; Performance IQ | Range: number of errors (maximum of 46) and years of education are inserted in three linear equations to calculate the three types of IQ |
| BDI-II[63 64] | A 21 question measure assessing the presence of depressive symptoms experienced by the participant within the past week. | Emotional functioning. | Range: 0–63 A cut-off score indicative of mild depressive symptoms is greater than 10 and for severe depressive symptoms is greater than 30. |
| Barthel ADL Index[65 66] | An index to measure functional disability, focused on bodily oriented personal care. | Functional domains: feeding; incontinence; transferring; toileting; dressing; bathing. | Range: 0–100 Lower scores reflect increased disability. |
| IADL[67 68] | An eight item scale used to assess independent living skills which include more complex activities (ie, 'instrumental activities of daily living') necessary for functioning in community settings. | Functional domains: using the telephone; shopping; food preparation; housekeeping; laundry; transport; medication; finances. | Range: 0–8 Higher scores reflect high function, independence. |

*Only patients who score below 1.5 SD of age-adjusted and education-adjusted cut-offs at the Montreal Cognitive Assessment during each evaluation (baseline, and 1, 3, 5, 7 and 10 years of follow-up) and a random sample of 30 participants will be invited for a neuropsychological evaluation.
Barthel ADL Index, Barthel Activities of Daily Living Index; BDI-II, Beck Depression Inventory-Second Edition; CDT, Clock Drawing Test; IADL, Lawton and Brody Instrumental Activities of Daily Living; SDMT, Symbol and Digit Modalities Test; SMC, Subjective Memory Complains scale; TeLPI, Irregular Word Reading Test; TMT, Trail Making Test; WAIS-III, Wechsler Adult Intelligence Scale-Third Edition; WMS-III, Wechsler Memory Scale-Third Edition.

domains, including attention, memory, executive functions, language, calculation, constructive capacity and visuospatial processing, through 11 exercises designed to include random elements and alternate sequences to lower the learning effect of repeating cognitive tests. It is to be performed using a home computer to access a web platform where different cognitive tests are uploaded. Each patient's results are stored and can be monitored by the research team.

Patients will be eligible to participate if they have completed at least 3 years of schooling, have no severe motor, visual or language impairments that prevent cognitive assessment, have easy access to a computer with an internet connection, and are able to use a computer without help. At the end of the baseline evaluation, participants will undergo a training session, and will be instructed to remotely log into the web platform and proceed to the first evaluation after 1 week, and then every 3 months. A Short Message Service reminder will be sent to participants 1 day before each remote evaluation. The research team will be automatically notified when participants fail to perform the test in order to reschedule the evaluation.

### Comprehensive neuropsychological assessment and clinical evaluation by a neurologist

All participants who score below 1.5 SD of age-adjusted and education-adjusted cut-offs on the Montreal Cognitive Assessment[31 32] in each evaluation (baseline, and 1, 3, 5, 7 and 10 years of follow-up) will undergo a

neuropsychological evaluation, expectedly within 1 month, comprising a battery of cognitive tests (table 2). The type and progressive nature of cognitive impairment, and its functional impact will be determined through a clinical evaluation performed by a neurologist, with a close surrogate present. Additionally, participants with a first neuropsychological evaluation will be reassessed with the same battery of tests, independently of their Montreal Cognitive Assessment score in the subsequent follow-up evaluations. A random sample of 30 patients with normal scores on the Montreal Cognitive Assessment will also perform a neuropsychological evaluation at 1, 3, 5, 7 and 10 years of follow-up, as a control group.

Participants will be classified as having mild cognitive impairment when presenting cognitive complaints over a period of at least 6 months, as reported by the patient or family members, modest cognitive decline from a previous level of performance reported by the patient or family members, and neuropsychological evaluation scores at least 2.0 SD below the age-corrected norms in at least one cognitive domain or at least 1.5 SD below the age-corrected norms in at least two cognitive domains, while also presenting no clinical depression or interference of cognitive function with independence in daily activities.[37 38]

Dementia will be defined according to the criteria used for defining major neurocognitive disorder of the Diagnostic and Statistical Manual of Mental Disorders (fifth edition), that is, significant cognitive impairment in at least one cognitive domain representing a significant decline from a previous level of functioning that interferes with independence in daily activities.[38] The severity of dementia will be classified using the clinical dementia rating scale.[39] The initial clinical classification will be confirmed after at least 6 months of clinical follow-up by a neurologist, and a complete diagnostic workup to identify other potential causes of cognitive impairment not related to oncological disease, including blood analyses for treatable causes of dementia and imaging studies.

## Data analyses and sample size

The frequency of cognitive decline and impairment will be described in the different categories of sociodemographic (age, education, employment, marital status) and lifestyle (alcohol intake, tobacco smoking, physical activity, and fruit and vegetables consumption) variables, as well as patient reported outcomes (anxiety, depression, quality of sleep), and according to the clinical characteristics of prostate cancer (cancer stage, risk strata) as well as treatments.

Trajectories of cognitive decline will be described through indicators of cognitive performance at different moments of evaluation using the appropriate format according to the nature of the variables and their distribution. Fixed-effects and mixed-effects models will be computed to compare cognitive performance trajectories (considering age and education) over time, according to

other sociodemographic and clinical characteristics, for each of the treatment groups.

Prevalence at baseline and incidence measures (incidence rates and cumulative incidences) and the corresponding 95% CIs will be estimated to quantify the frequency of cognitive impairment, and the association between treatments and incident cognitive impairment. Cumulative incidence will be estimated considering death as a competing event, according to the Kalbfleisch and Prentice method.[40] Crude and adjusted relative risks will be calculated.

The sample size was calculated considering the objective to quantify the association between the use of androgen deprivation therapy and cognitive decline between the baseline and the 1-year evaluation, defined as a variation in the score from baseline to the 1-year evaluation below 1.5 SD of the distribution in the cohort, of the changes in cognitive scores over the same time period. For this, assuming a statistical power of 80%, a level of significance of 5% and a 1:1 ratio between androgen deprivation therapy-exposed (groups 3 and 4) and unexposed (groups 1 and 2), 600 prostate cancer patients will be necessary to detect a twofold higher proportion of participants (14%) with cognitive decline in the androgen deprivation therapy group. Secondary analyses will be conducted considering the exposure to each specific hormonal treatment.

For the description of cognitive performance trajectories, and the calculation of the prevalence of cognitive impairment at baseline and incidence measures (incidence rates and cumulative incidences), the sample size will influence the precision of the estimates at each moment but will not be a limiting factor for the essentially descriptive accomplishment of these objectives. Nevertheless, considering the prevalence of cognitive impairment in the general population of Northern Portugal of 9.6%,[41] a precision of 2.4%, and a 95% confidence level, a sample of 579 individuals will be needed. As such, the estimated sample size calculated above will also be sufficient for estimating the prevalence of cognitive impairment in the population of patients with prostate cancer.

Considering the high potential for confounding by indication, propensity scores calculated based on several disease characteristics, including prognostic biomarkers and predictors of response to treatment, will be used in data analysis. Causal diagrams will be used to support the decisions regarding the potential role of the different sociodemographic, lifestyle, clinical and treatment variables in the causal pathways.

Training of interviewers and the use of standardised procedures for data collection are expected to contribute to a low proportion of missing data, and no imputation is being planned.

Considering our experience in another cancer cohort,[42] we estimate that approximately a third of the total sample will participate in the Brain on Track evaluation. Using as criteria for referral of participants to the comprehensive neuropsychological assessment, the Montreal Cognitive

Assessment cut-off score, 1.5 SD below the mean of age and education-based group distribution from published normative data,[32] we expect at least 42 patients to undergo a neuropsychological assessment at baseline and at each subsequent evaluation.

Taking into account the survival of patients with prostate cancer in the North of Portugal,[43] and the high participation obtained in a previous prospective cohort study of patients with breast cancer;[42 44] we estimate at least 90% and 80% of patients will participate in the 1-year and 5-year follow-up evaluations, respectively. In order to minimise refusals and losses to follow-up, all evaluations will be scheduled to take place on the same day as routine appointments in the respective hospital and participants will be invited again when they miss scheduled appointments.

## Contingency plan

Due to the COVID-19 pandemic, recruitment and evaluation of participants were interrupted from March to June 2020. Beginning in July, procedures were adapted to minimise the risk of infection for participants and members of the research team. Only the Montreal Cognitive Assessment and the neuropsychological evaluation will be performed face-to-face at the hospital. Participants will answer the questionnaire on sociodemographic, and lifestyle and dietary characteristics during a telephone interview. Self-administered questionnaires will be completed at home and sent mailed back with a prepaid envelope.

Anthropometrics measurements, blood sample collection and the Cube Test evaluation will not be performed. Weight, height, blood pressure and blood sample parameters will be retrieved from medical records when available or asked to the participants. The initial training session for the Brain on Track evaluation will be conducted through videoconference.

The impact of the pandemic on the course of this investigation, namely regarding participation and retention rates, completeness of information and potential losses of validity and precision will be addressed specifically. Additional mitigation measures may have to be adopted, namely an extension of the recruitment period or an increase in the sample size.

## Patient and public involvement

Patients and public were not involved in the conception, design and dissemination of this study.

## ETHICS AND DISSEMINATION

Ethics approval was obtained from the Ethics Committees of the Portuguese Institute of Oncology of Porto (Ref. CES 89/017) and the São João Hospital Centre (Ref. 76/17), and by the Portuguese Data Protection Authority (Authorisation 3478/2017). Written informed consent will be obtained from all participants after the project's aims and procedures are fully explained by a member of the research team.

This is an observational study in which patients with prostate cancer will be followed according to usual clinical practice, as such the occurrence of harmful effects related to participation in the study are not expected. Participants will receive detailed information about the research purpose and objectives, name and institution of the researchers, expected duration of the interview, voluntary nature of participation, clearly stating that there will be no penalty for those who refuse to participate, and ensuring confidentiality and anonymity of all the information provided. Participants will be asked to give authorisation for collection of data from their personal clinical records. After clarification of any doubts, an informed consent will be signed in duplicate and a copy will be given to each participant. All participants will be informed that they can leave the study at any time, and this decision does not affect their medical care. There is no expected risk or discomfort other than those arising from interviewing, collecting venous blood samples and physical measurements (height, weight, blood pressure). Only participants able to understand the study and provide informed consent will be included. To minimise possible discomfort due to the required trips to the hospital for face-to-face evaluations or the duration of interviews, and to avoid unnecessary burden and travel expenses, data collection procedures were designed to last no more than 60 min, and will be scheduled to take place on the same day as other appointments in the respective hospitals as part of regular clinical care, preferably in the morning due to the fasting requirement. Starting in July 2020, only the Montreal Cognitive Assessment will be performed face-to-face at the hospital, to reduce the risk of infection by SARS-CoV-2.

This study requires the collection and processing of sensitive personal data including health and clinical data from questionnaires and the clinical files of patients. Therefore, additional measures will be taken to protect the anonymity and the confidentiality of all participants. All data regarding clinical aspects will be collected by clinical members of the research team and privacy is assured. All participants will have a study-specific identification number, which will be used in all questionnaires and stored blood samples. The correspondence between this identification number and the personal identifiable information will be stored in a file, to which only the principal investigator will have access. Only the research team will have access to the database with anonymised data, saved on a password-protected secure computer. No personal identifiers will be used in data analyses. The same procedures will be adopted for each of the evaluations.

The expected results may contribute to elucidate the magnitude of the androgen deprivation therapy effect on the cognitive function of patients with prostate cancer, and the possible mediator effect of metabolic syndrome and anaemia in this process. This may help clinical decisions regarding the pharmacological class to be used in patients more vulnerable to cognitive impairment. This study may also contribute to the refinement and

validation of the longitudinal monitoring tool Brain on Track. Considering the 10-year temporal horizon of this project, the follow-up of the cohort assembled will contribute to a better understanding of the long-term trajectories of cognitive performance and the iatrogenic effects of prostate cancer treatments.

The findings of this project will be submitted for publication in international peer-reviewed journals, and proposed for presentation in relevant national and international conferences, which will allow for the dissemination of the main findings across the medical community. Press releases through mass media will also be issued to promote the dissemination of information relevant to the general population and policy-makers. Furthermore, the project will contribute to the training of researchers through the production of masters' theses and doctoral dissertations.

**Author affiliations**
[1]EPIUnit, Instituto de Saúde Pública da Universidade do Porto, Porto, Portugal
[2]Departamento de Ciências da Saúde Pública e Forenses e Educação Médica, Faculdade de Medicina da Universidade do Porto, Porto, Portugal
[3]USF Lagoa, Unidade Local de Saúde de Matosinhos EPE, Senhora da Hora, Portugal
[4]Serviço de Oncologia, Instituto Português de Oncologia do Porto Francisco Gentil, EPE, Porto, Portugal
[5]Serviço de Neurologia, Unidade Local de Saúde de Matosinhos EPE, Senhora da Hora, Portugal
[6]Serviço de Neurologia, Centro Hospitalar de Entre o Douro e Vouga EPE, Santa Maria da Feira, Portugal
[7]Serviço de Urologia, Instituto Português de Oncologia do Porto Francisco Gentil, EPE, Porto, Portugal
[8]Instituto de Investigação em Ciências da Vida e Saúde, Escola de Medicina da Universidade do Minho, Braga, Portugal
[9]Serviço de Urologia, Centro Hospitalar de São João EPE, Porto, Portugal
[10]Serviço de Neurologia, Instituto Português de Oncologia do Porto Francisco Gentil, EPE, Porto, Portugal

**Contributors** NL and SP conceived and designed the study. NA wrote the first version of the manuscript. ARC, AFC, JO, LPF, LR, NL, RB, SM, SP and VTC critically revised the manuscript for relevant intellectual content. All authors approved the final version for submission.

**Funding** This study was funded by FEDER through the Operational Programme Competitiveness and Internationalisation and national funding from the Foundation for Science and Technology-FCT (Portuguese Ministry of Science, Technology and Higher Education) under the project 'NEON-PC - Neuro-oncological complications of prostate cancer: longitudinal study of cognitive decline' (POCI-01-0145-FEDER-032358; Ref. PTDC/SAU-EPI/32358/2017), and the Unidade de Investigação em Epidemiologia - Instituto de Saúde Pública da Universidade do Porto (EPIUnit) (UIDB/04750/2020) financed by national funds from FCT. SM was funded under the scope of the project 'NEON-PC - Neuro-oncological complications of prostate cancer: longitudinal study of cognitive decline' (POCI-01-0145-FEDER-032358; Ref. PTDC/SAU-EPI/32358/2017). Individual PhD grants attributed to ARC (SFRH/BD/102181/2014) and NA (SFRH/BD/119390/2016) were funded by FCT and the 'Programa Operacional Capital Humano' (POCH/FSE).

**Competing interests** VTC has a shareholder position in Neuroinova, Lda a start-up company that conceived Brain on Track, holds registered trademark and commercialization rights.

**Patient and public involvement** Patients and/or the public were not involved in the design, or conduct, or reporting, or dissemination plans of this research.

**Patient consent for publication** Not required.

**Provenance and peer review** Not commissioned; externally peer reviewed.

**ORCID iD**
Nuno Lunet http://orcid.org/0000-0003-1870-1430

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
