## [Reviewer comments · BMJ Open]

ARTICLE DETAILS

TITLE (PROVISIONAL)	Cognitive decline in prostate cancer patients: study protocol of a prospective cohort – NEON-PC
AUTHORS	Araujo, Natalia; Morais, Samantha; Costa, Ana; Braga, Raquel; Carneiro, Ana; Cruz, Vitor; Ruano, Luis; Oliveira, Jorge; Figueiredo, Luis; Pereira, Susana; Lunet, Nuno

VERSION 1 – REVIEW

REVIEWER	Sonia García-Cabezas Reina Sofia University Hospital, Cordoba, Spain.
REVIEW RETURNED	20-Sep-2020

GENERAL COMMENTS	This is an interesting project to study the effect of androgen deprivation (direct and indirect) on the cognitive function of patients with prostate cancer undergoing this treatment. This effect has already been reported in previous studies, but this project aims to make up for the shortcomings of previous studies (difficult to compare), with a more adequate methodology and design. Its sample size and the 10-year follow-up period will provide more reliable and realistic results. Prostate cancer is increasingly prevalent and it is important to know the effects that androgen deprivation can have on the quality of life of these patients. The cognitive and general evaluation of patients is exhaustively assessed by means of a large number of questionnaires, instruments and scales that cover a great deal of necessary information in different aspects and domains, as well as the assessment by a neurologist if a cognitive alteration is detected. The objectives of the project also include the evaluation of the metabolic syndrome and anemia as possible mediators of the effect of androgen deprivation on cognitive function. In my opinion it is a well justified and relevant work. However, I wanted to make a few comments: -Groups 1 and 2 do not include androgen blockade treatment. Are they considered as controls?-The groups that include chemotherapy may present greater deterioration, both cognitive and analytical, for example increased anemia. In addition, life expectancy in this subgroup of patients is limited, since they would already be metastatic patients, so the follow-up will be considerably shorter.-I imagine that different molecules will be allowed for androgen blockade (not specified) and that the duration of treatment will depend on the risk and staging of the patient.-The blood sample to be taken does not indicate that testosterone will be determined.
--

	-Has it been contemplated that a proportion of patients in the study will probably receive, in the event of progression, treatment with new antiandrogens (Abiraterone / enzalutamide)? Enzalutamide has also been associated with cognitive decline. -In its contingency plan for the COVID-19 pandemic it is stated that: "... blood sample collection... will not be performed". "... Blood sample parameters will be retrieved from medical records when available ...". This will detract from the third objective: "To assess the role of metabolic syndrome and anemia as possible mediators of the androgen deprivation therapy effect on cognitive performance". Has the recruitment period been extended when the pandemic improves?
--	--

REVIEWER	Bernardo Cacho-Díaz INCAN (National Cancer Institute) Mexico, Mexico City
REVIEW RETURNED	21-Sep-2020

GENERAL COMMENTS	It is a very interesting and relevant study. The investigators have already started recruitment, but during the text they mention some procedures will be done and some have already been done, they should clarify the phase of the study at the present time. For example: "We describe a prospective cohort study that will evaluate prostate cancer patients selected among those being treated at the two largest hospitals providing cancer care in the North of Portugal, which attend half the prostate cancer patients in this region". In the paper reviewed, no description is included. They clearly explain the objectives but, at this point, a hypothesis should be stated for many purposes: to correctly calculate the sample size, to establish a one tail or two tail approach of the statistic procedures, to establish a magnitude and direction of the study. It is not clear in the text if they want to study the frequency of cognitive decline or if there is an effect of the treatment in cognitive functioning. In the title they mention all Neuro-oncologic manifestations, but they are only studying the cognitive impact of androgen deprivation therapy, a correction of the title is pertinent. There are some bias that should be taken into account:  - How are they going to differentiate between cognitive decline due to androgen deprivation vs other metabolic, reversible, vascular, metastatic, inflammatory, degenerative and other causes? - Cognitive decline should be adjusted for age, scholary and other cofounding variables, how are they addressing this? For it is not explained or described. - What test are they going to use as a standard for the diagnosis of cognitive decline? A standard complete neuropsychologic and neurologic evaluation should be added for MOCA test is a very good screening test but a confirmatory or standard complete evaluation should be encouraged. - In order to diagnose cognitive decline, delirium and other psychiatric conditions should be excluded, how are they going to do that?
--

VERSION 1 – AUTHOR RESPONSE

REVIEWER #1

Comment #1

This is an interesting project to study the effect of androgen deprivation (direct and indirect) on the cognitive function of patients with prostate cancer undergoing this treatment.

This effect has already been reported in previous studies, but this project aims to make up for the shortcomings of previous studies (difficult to compare), with a more adequate methodology and design. Its sample size and the 10-year follow-up period will provide more reliable and realistic results.

Prostate cancer is increasingly prevalent and it is important to know the effects that androgen deprivation can have on the quality of life of these patients.

The cognitive and general evaluation of patients is exhaustively assessed by means of a large number of questionnaires, instruments and scales that cover a great deal of necessary information in different aspects and domains, as well as the assessment by a neurologist if a cognitive alteration is detected.

The objectives of the project also include the evaluation of the metabolic syndrome and anemia as possible mediators of the effect of androgen deprivation on cognitive function.

In my opinion it is a well justified and relevant work.

Reply to comment #1

We thank the reviewer for these favourable comments to our manuscript. No specific question was raised.

Comment #2

However, I wanted to make a few comments:

-Groups 1 and 2 do not include androgen blockade treatment. Are they considered as controls?

Reply to comment #2

To evaluate the association between ADT exposure and cognitive decline, groups 1 and 2 will be considered as reference groups or controls, as they are not exposed to ADT. We now make this clearer in the manuscript (please see page 8, paragraph 4).

Comment #3

The groups that include chemotherapy may present greater deterioration, both cognitive and analytical, for example increased anemia. In addition, life expectancy in this subgroup of patients is limited, since they would already be metastatic patients, so the follow-up will be considerably shorter.

Reply to comment #3

We recognize that the group of patients treated with chemotherapy may have a worse health state and a shorter follow-up time.

In the revised manuscript, we clarify that variables that are potential confounders (e.g. anemia at baseline) or mediators of effects (e.g. incident anemia during follow-up) will be identified using causal diagrams and taken into account in data analysis accordingly (please see page 9 paragraph 2).

For the analytical objective of comparing the occurrence of cognitive decline during the first year of follow up between ADT-exposed and ADT-non-exposed participants, we believe the follow-up time will not be a bias. For longer follow-up time, we will consider death as a competitive event for the calculation of risk ratios. The role of death as a competing event and the need to take that into

account when comparing groups with differing survival is acknowledged in the data analysis section of the manuscript (please see page 8 , paragraph 3).

Comment #4

I imagine that different molecules will be allowed for androgen blockade (not specified) and that the duration of treatment will depend on the risk and staging of the patient.

Reply to comment #4

We will consider all drugs used for systemic treatment of prostate cancer as well as the duration of treatment; the manuscript was revised to make this clearer (please see page 3, paragraph 4). Considering that the type of treatment depends on cancer stage and risk, we will use propensity scores (the probability of treatment assignment conditional on observed baseline characteristics) to reduce confounding by indication. Cancer staging will be defined based on the AJCC TNM 8th edition and risk stratification according to the National Comprehensive Cancer Network (www.nccn.org); this is also acknowledged in the revised manuscript (please see page 3, paragraph 4 and page 9, paragraph 2).

Comment #5

The blood sample to be taken does not indicate that testosterone will be determined.

Reply to comment #5

Blood samples will not be used to determine testosterone. This biomarker is important for clinical follow-up of patients under ADT, and may be obtained from clinical records. Castration level of testosterone are thought to be a direct mechanism of the negative effect of ADT on cognitive performance, and post-treatment levels of testosterone are an intermediate step between treatment and cognitive decline that may be mediated by testosterone.

We believe that information on testosterone pre-treatment levels is not essential to control for its potential confounding effect, since such a confounding effect may be accounted for in data analysis by controlling for other variables in the same pathway. For example, a potential confounding effect of pre-treatment testosterone levels due to the effect of obesity on both testosterone and treatment options may be accounted for by conditioning for the obesity variable.

Comment #6

Has it been contemplated that a proportion of patients in the study will probably receive, in the event of progression, treatment with new antiandrogens (Abiraterone / enzalutamide)? Enzalutamide has also been associated with cognitive decline.

Reply to comment #6

This is an observational study and patients will be clinically followed according to usual practice, which may include new antiandrogens and additional treatments after initial therapy. We now clarify in the manuscript that data regarding all drugs used for systemic treatment of prostate cancer, either at initial or follow-up treatments, will be collected (please see page 3, paragraph 4). In fact, it is expected that most participants initiating ADT will be administered goserelin, and in fewer cases leuprorelin, triptorelin, or degarelix. Bicalutamide in combination with an LHRH agonist may also be used in some cases. The new antiandrogens, enzalutamide and abiraterone, and the chemotherapy drug docetaxel are expected to be administered less frequently in the first year of follow-up as these drugs are prescribed if specific criteria are met, according to evidence of recent trials. Immunotherapy for prostate cancer is not currently being used at IPO nor at CHSJ.

Secondary analyses will be conducted considering the exposure to each specific hormonal treatment, and the role of each variable in the causal pathways will be decided upon the analysis of causal

diagrams. This information is now being included in the revised version of the manuscript (please see page 8, paragraph 4 and page 9, paragraph 2).

Comment #7

-In its contingency plan for the COVID-19 pandemic it is stated that: "... blood sample collection... will not be performed". "... Blood sample parameters will be retrieved from medical records when available ...". This will detract from the third objective: "To assess the role of metabolic syndrome and anemia as possible mediators of the androgen deprivation therapy effect on cognitive performance". Has the recruitment period been extended when the pandemic improves?

Reply to comment #7

Recruitment was established to occur until sample size was reached. In March 2020, field activities, namely recruitment, had to be suspended. In July 2020, we were authorized to resume evaluations, including recruitment, with the condition of limiting physical contact with participants. The COVID-19 pandemic has been affecting both baseline and follow-up evaluations, either regarding their timing and the quality of the variables collected, namely hematologic and biochemical parameters, and anthropometric measurements.

Alternative procedures to collect these variables have been adopted, namely for serum hemoglobin, using laboratory results from electronic files, as this parameter is routinely assessed at IPO, at least before and during treatment. Blood pressure measurements are also expected to be performed routinely by nurses before treatments. In participants taking medications to lower cholesterol, reporting taking these medicines can substitute the information of a high level of cholesterol, but lipid profile in participants not taking cholesterol lowering drugs, is essential to assess the presence of metabolic syndrome. Unfortunately, we expect many regular check-ups with family doctors have been postponed and recent laboratory results may not be available. Lipid profile provides two of the five criteria usually used to define metabolic syndrome. Cognitive evaluations, however, are being conducted in person (or remotely for the web-based cognitive instrument Brain on Track) as they were performed before the pandemic.

Although participation and retention rates may decrease, and missing data is expected to increase, namely regarding the variables needed to define metabolic syndrome, we believe the contingency measures adopted will minimize the impacts of the pandemic in our study. However, in the revised version of the manuscript we now acknowledge that the impact of the pandemic on the course of this investigation will need to be addressed specifically, namely regarding participation and retention rates, completeness of information and potential losses of validity and precision, and that additional mitigation measures may have to be adopted, namely an extension of the recruitment period or an increase in the sample size (please see page 10, paragraph 3).

REVIEWER #2

Comment #1

It is a very interesting and relevant study.

The investigators have already started recruitment, but during the text they mention some procedures will be done and some have already been done, they should clarify the phase of the study at the present time.

For example: "We describe a prospective cohort study that will evaluate prostate cancer patients selected among those being treated at the two largest hospitals providing cancer care in the North of Portugal, which attend half the prostate cancer patients in this region". In the paper reviewed, no description is included.

Reply to comment #1

We are now making clear in the main text of the manuscript that recruitment started in February, 2018 and is ongoing, and that is expected to be completed in the first semester of 2021 (please see page 2, paragraph 3).

Comment #2

They clearly explain the objectives but, at this point, a hypothesis should be stated for many purposes: to correctly calculate the sample size, to establish a one tail or two tail approach of the statistic procedures, to establish a magnitude and direction of the study. It is not clear in the text if they want to study the frequency of cognitive decline or if there is an effect of the treatment in cognitive functioning.

Reply to comment #2

The hypothesis is that treatments with ADT for prostate cancer can cause cognitive deterioration, which may be a benign decrease in cognitive performance, or a pathologic condition such as cognitive impairment or dementia. As cognitive decline precedes cognitive impairment and dementia, we test the hypothesis, using the short-term cognitive outcome, decrease in cognitive performance after one year of follow-up. Considering the possibility of practice effects, we defined the outcome of decrease in cognitive performance as a variation in the MoCA score below 1.5 standard deviation of the variation observed in the cohort. Sample size calculation was based on the comparison of the frequency of this outcome in ADT-exposed and ADT-non-exposed participants, and setting a two-fold higher frequency in the former in relation to the latter. We revised the sample size and data analysis section to make this clearer (please see page 8, paragraph 4).

Besides this analytical objective, we aim to quantify the burden of cognitive impairment in prostate cancer patients before and after different treatments, by estimating its frequency of occurrence. Finally, we aim to identify and describe trajectories of cognitive performance over time using MoCA and Brain on Track. This approach allows us to assess the acute or progressive nature of cognitive deterioration, and of its reversibility/persistence.

Comment #3

In the title they mention all Neuro-oncologic manifestations, but they are only studying the cognitive impact of androgen deprivation therapy, a correction of the title is pertinent.

Reply to comment #3

We recognize that the first part of the title may seem too broad for the real objective of studying cognitive decline. We changed the title to "Cognitive decline in prostate cancer patients: study protocol of a prospective cohort – NEON-PC"

Comment #4

There are some bias that should be taken into account:

- How are they going to differentiate between cognitive decline due to androgen deprivation vs other metabolic, reversible, vascular, metastatic, inflammatory, degenerative and other causes?

Reply to comment #4

Anemia, metabolic syndrome, co-morbidities and cancer stage, that are indicative of metabolic, vascular, metastatic and inflammatory causes of cognitive deterioration will be assessed at baseline and will be introduced in the statistics models (logistic regression and survival analyses) as confounders of the association between androgen deprivation therapy and cognitive decline. The etiology of cognitive decline (a decrease in the MoCA score below the normal range of the distribution of the variation in MoCA observed in the cohort) will only be studied if the MoCA score

suggests cognitive impairment. In this case, neuropsychological tests and a clinical evaluation by a neurologist will confirm cognitive impairment, and laboratory tests and imaging exams will be used to complement the study of its etiology.

In the revised manuscript, we now specify that causal diagrams will be used to support the decisions regarding the potential role of the different sociodemographic, lifestyle, clinical and treatment variables in the causal pathways (please see page 9, paragraph 2). This will allow for the distinction between variables that are confounders from those that have a mediator role in the causal pathways.

Comment #5

- Cognitive decline should be adjusted for age, scholary and other cofounding variables, how are they addressing this? For it is not explained or described.

Reply to comment #5

The age- and education-adjusted cut-offs of MoCA will be used (please see page 5, paragraph 4). The frequency of cognitive decline and impairment will be described in the different categories of sociodemographic (age, education, employment, marital status) and lifestyle (alcohol intake, tobacco smoking, physical activity, and fruit and vegetables consumption) variables, as well as patient reported outcomes (anxiety, depression, quality of sleep), and according to the clinical characteristics of prostate cancer (cancer stage, risk strata) as well as treatments. This was added to the revised version of the manuscript (please see page 7, paragraph 4).

To test the hypothesis of a two-fold more likelihood of cognitive decline in ADT-exposed versus ADT-non-exposed patients, we will compute crude and adjusted odds ratio and their 95% confidence intervals, introducing the above mentioned variables in logistic regression models. In the Data analyses and sample size section of the manuscript we refer: "Fixed- and mixed-effects models will be computed to compare cognitive performance trajectories (considering age and education) over time, according to other sociodemographic and clinical characteristics, for each of the treatment groups." and "Crude and adjusted relative risks will be calculated." (please see page 8, paragraphs 2 and 3).

Comment #6

- What test are they going to use as a standard for the diagnosis of cognitive decline? A standard complete neuropsychologic and neurologic evaluation should be added for MOCA test is a very good screening test but a confirmatory or standard complete evaluation should be encouraged.

Reply to comment #6

In the section "Comprehensive neuropsychological assessment and clinical evaluation by a neurologist" we describe the procedures for the complete cognitive assessment of the participants: "All participants who score below 1.5 standard deviations of age- and education-adjusted cut-offs on the Montreal Cognitive Assessment in each evaluation (baseline, and one, three, five, seven and 10 years of follow-up) will undergo a neuropsychological evaluation, expectedly within one-month, comprising a battery of cognitive tests (Table 2). The type and progressive nature of cognitive impairment, and its functional impact will be determined through a clinical evaluation performed by a neurologist, with a close surrogate present" (please see page 6, paragraph 4, and Table 2).

Comment #7

- In order to diagnose cognitive decline, delirium and other psychiatric conditions should be excluded, how are they going to do that?

Reply to comment #7

A suspected case of cognitive impairment detected using MoCA, will be examined in more detail with a neuropsychological battery of tests administered by an experienced neuropsychologist and by clinical evaluation by a neurologist (please see page 6, paragraph 4, and Table 2). Exclusion of delirium and other psychiatric conditions will also be performed after the clinical evaluation by a neurologist based on the criteria of the Diagnostic and Statistical Manual of Mental Disorders (fifth edition).

VERSION 2 – REVIEW

REVIEWER	Sonia García-Cabezas Reina Sofia University Hospital Spain
REVIEW RETURNED	28-Nov-2020
GENERAL COMMENTS	The authors have adequately answered the questions requested and clarified in the text different sections of the study in response to these questions.
REVIEWER	Cacho-Díaz, Bernardo MD MSc National Cancer Institute, Mexico
REVIEW RETURNED	25-Nov-2020
GENERAL COMMENTS	Authors have addressed all comments and observations